# The Deuterium Oxide Dilution Method to Quantify Human Milk Intake Volume of Infants: A Systematic Review—A Contribution from the ConcePTION Project

**DOI:** 10.3390/nu16234205

**Published:** 2024-12-05

**Authors:** Lucas Cloostermans, Karel Allegaert, Anne Smits, Martje Van Neste

**Affiliations:** 1Faculty of Medicine, KU Leuven, 3000 Leuven, Belgium; 2Clinical Pharmacology and Pharmacotherapy, Department of Pharmaceutical and Pharmacological Sciences, KU Leuven, 3000 Leuven, Belgium; 3L-C&Y, KU Leuven Child & Youth Institute, 3000 Leuven, Belgium; anne.smits@uzleuven.be; 4Department of Development and Regeneration, KU Leuven, 3000 Leuven, Belgium; 5Department of Hospital Pharmacy, Erasmus University Medical Center, 3000 CA Rotterdam, The Netherlands; 6Neonatal Intensive Care Unit, University Hospitals Leuven, 3000 Leuven, Belgium

**Keywords:** breastfeeding, lactation, infants, human milk intake volume, physiologically based pharmacokinetic (PBPK) modeling, deuterium oxide dilution method, stable isotope

## Abstract

**Background:** Global health organizations recommend breastfeeding, but maternal pharmacotherapy can disrupt this due to safety concerns. Physiologically based pharmacokinetic (PBPK) models predict medication transfer through breastfeeding, relying on validated milk intake volume data. However, the literature is mainly focused on different measurement methods, or such intake data have been collected without systematic review. This systematic review therefore aims to gather data on human milk intake volume derived using the (dose-to-the-mother) deuterium oxide dilution method, allowing for comparison with the literature. Additionally, it aims to explore the effects of maternal conditions on milk intake volume. **Methods:** PubMed, Embase, Web of science, Cochrane library, Scopus and CINAHL were searched for studies on the dilution method and breastfeeding in healthy infants. Risk of bias was assessed using the Newcastle–Ottawa scale (NOS) and the Risk of Bias 2 (RoB2) tool. Data on mean human milk intake volume were extracted and synthesized (mL/day and mL/kg/day) throughout infancy. **Results:** Sixty studies (34 countries) reported on the milk intake volume of 5502 infants. This intake was best described by logarithmic regression y(mL/kg/day) = 149.4002 − 0.2268 × x − 0.1365 × log(x) (x = postnatal age, days). Maternal conditions showed no significant influence on human milk intake, except for maternal smoking (reduction). **Conclusions:** This function corresponds with previous research, particularly for infants aged between 1.5 and 12 months. The limited availability of early infancy data underscores the need for additional data for future PBPK modeling to enhance informed healthcare decisions and improve outcomes for mothers and infants.

## 1. Introduction

The World Health Organization (WHO) emphasizes the importance of breastfeeding, recommending exclusive breastfeeding up to six months of age followed by partial breastfeeding until two years of age or longer [1]. This approach has been proven to be beneficial for the infant, e.g., reducing the risk of gastro-intestinal infections or leukemia in early life and lasting advantages later in life [1,2]. Breastfeeding also provides various health benefits to the mother, including a lower risk of ovarian and breast cancer [2]. From six months of age, breastfeeding should be complemented with additional food sources, as the nutritional needs of the growing infant are no longer adequately met through exclusive breastfeeding alone [1]. Despite these WHO recommendations, not all women breastfeed their infant for as long as they desire or intend [3]. Maternal conditions and the resulting decision to start medication can be a reason for preliminary termination of breastfeeding [3]. Furthermore, healthcare workers may hesitate to prescribe pharmacotherapy or advise mothers to (temporarily) interrupt breastfeeding because of the limited understanding of the safety of maternal medication for the nursing infant [4].

To inform shared decision making among patients and healthcare workers, insights into the extent and safety of medication transferred from mother to infant through human milk are necessary. Clinical lactation studies can provide these insights, but they are commonly costly, logistically challenging and ethically complex [5]. Therefore, there is room for physiologically based pharmacokinetic (PBPK) models to add to the available evidence [5]. These mathematical models merge physiological population-specific and medication-specific knowledge to predict medication concentrations [5,6]. In this manner, infantile drug exposure can be simulated by multiplying the concentration of medication in human milk with the milk intake volume of the infant [5,7].

Mother–infant PBPK models rely on accurate physiological input data, which are frequently lacking [5,6]. Among these variables, knowledge about the human milk intake volume of infants is crucial [5,7]. The two most widely used techniques to quantify this variable are the (dose-to-the mother) deuterium oxide dilution method and the test-weighing method [8,9]. In the 14-day deuterium oxide dilution method, a small and deemed safe quantity of deuterium oxide (e.g., 30 g of 99.8% ^2^H) is orally administered to the mother and dispersed to the infant through breastfeeding [8]. This method then imputes the measured infantile and maternal deuterium concentration in saliva (or urine) collected on day 1, 2, 3, 4, 13 and 14 into a theoretical model of water turnover in the mother–infant entity. This way the method can estimate human milk intake volume over the course of two weeks [8]. Differently, the test-weighing method compares infantile weight before and after feeding to quantify milk intake [9]. However, this method is prone to systematic errors when improper scales or procedures are selected [10]. The limitations of this method are possibly more present in the first weeks of postnatal age, where human milk intake volume quantities are low [11]. In contrast, the deuterium oxide dilution method has been found to be especially effective in infants up to six months of age without disturbing normal feeding behavior [8,9]. Accurate assessment of infant feeding is relevant for clinical practice and research. Based on 32 studies that evaluated the validity of direct observation, test weighing, or radiolabeled techniques, correlations with validation standards were highest for doubly labeled water and test weighing, and lowest for observation. However, there are cost and availability issues related to radiolabeled techniques like the deuterium technique [12].

Previously published research often combines data from the test-weighing and deuterium oxide dilution methods or is rather based on comprehensive reviews [7,13,14,15]. We are unaware of a systematic review on the available data of the deuterium oxide dilution method.

Consequently, we aimed to systematically collect available data from clinical studies that report human milk volumes solely using the deuterium oxide dilution method at different postnatal ages. These data are used to develop a function describing human milk intake volume over the postnatal period and thus allowing for a comparison with previously described functions by Yeung et al. or Rios-Leyvraz et al. [7,15].

Additionally, we analyzed the effects of specific maternal conditions (e.g., human immunodeficiency virus (HIV) status, morbidity characteristics, smoking) on the human milk intake volume of healthy infants. Hereby, this review can provide valuable input for mother–infant PBPK models, contributing to a deeper understanding of human milk intake volume at different infancy stages. Ultimately, this can advance infant medication safety during breastfeeding, enabling shared healthcare decisions and improved outcomes for both mother and infant.

## 2. Materials and Methods

This systematic review was registered in the International Prospective Register of Systematic Reviews (PROSPERO, CRD42022380374) [16]. This review was conducted to assess published scientific research on human milk intake volume in accordance with the Preferred Reporting Items for Systematic Reviews and Meta Analyses (PRISMA) guidelines (see Appendix A) [17]. Appendix A on search strategy, quality assessment, pooling formula, and data extraction and synthesis are available.

### 2.1. Inclusion and Exclusion Criteria

Eligible articles were primary studies reporting human milk intake volume using the dose-to-mother deuterium oxide dilution method in breastfed infants up to one year of age. The data on exclusive breastfeeding up to six months of age were of primary interest.

Available articles in English were included without restrictions on publication date, country of recruited participants or ethnicity. Conference abstracts were excluded in our systematic review. Additionally, infants recruited with medical conditions affecting human milk intake volume were excluded based on the definition of (un)healthy infants as applied in the retained studies. There were no limitations on studies reporting the incidental sickness of infants during the study, nor was there a limitation on maternal medical conditions or maternal weight. Preterm infants were excluded, with the used definition of being born prior to 36 weeks of completed gestation (<36 weeks gestational age) to accommodate divergent global definitions. There was no limitation on infantile weight, except for studies only including infants that were small for gestational age, as these were excluded. These criteria were separately evaluated for both control and intervention groups in controlled trials.

### 2.2. Search Strategy

The search strategy was developed in consultation with the Biomedical Sciences Library of KU Leuven (Leuven, Belgium). Six scientific databases were searched on 15 April 2023, i.e., PubMed, Embase, Web of science, Cochrane library, Scopus and CINAHL. No unpublished studies or gray literature were searched. The search string (Appendix A) consisted of two concepts that were subsequently merged, namely the deuterium oxide dilution method and breastfeeding.

### 2.3. Screening Process and Quality Assessment

Articles derived from the databases using the search string were imported into Covidence, a review management software tool [18]. Duplications were automatically removed and subsequently checked manually by one researcher (L.C.). In the first phase, the title and abstract were screened in accordance with the eligibility criteria by two researchers (L.C., M.V.N.) working separately. In the second phase, the full text of the selected articles was separately reviewed using the same methodology. In the case of any disagreement during either phase, a third researcher (K.A.) was consulted for the final and independent decision.

Quality assessment was performed by two researchers (L.C., K.A.), with the involvement of a third researcher (M.V.N.) in case of disagreement, using the adapted Newcastle–Ottawa scale (NOS) for cohort and cross-sectional studies (Appendix A) [19]. Quality was assessed regarding the used deuterium oxide dilution method, where cohort studies were defined as using the method in different cohorts and preferably using repeated measures within cohorts. The Risk of Bias (ROB) 2 tool was used for bias assessment of the randomized controlled trials (Appendix A) [20].

### 2.4. Data Extraction and Synthesis

Data extraction was performed using a structured extraction form in Microsoft Excel (version 16.81). Extraction was performed by one researcher (K.A., L.C.) with subsequent verification by a second researcher (M.V.N. or K.A., respectively). Data of primary interest included the human milk intake volume and infantile weight with their sample sizes at the reported postnatal age. The secondary data of interest encompassed variables such as the country of study, maternal weight, age, conditions, infantile gestational age and birth weight.

During synthesis, data were retained once per cohort and deuterium measurement period by merging articles on likely shared cohorts. We pooled secondary variables to describe the mother–infant pairs as accurately as possible (pooling formula in Appendix A) [21,22]. Standard errors and confidence intervals were converted to standard deviations (SDs) [23]. In the event that human milk intake volume (mean ± SD) was not reported in mL/day or mL/kg/day, values were converted from g/day or g/kg/day using the density of breast milk (1.03 g/mL) [7]. Additionally, the mean human milk intake volume was converted from mL/day to mL/kg/day using the given mean infant weight. To estimate the SD of these newly calculated means, the propagation of uncertainty principle was used, assuming the independence of errors between weight and human milk intake volume [24]. If the postnatal age of infants was unavailable but the day of the deuterium dose could be inferred (from the text, figures or tables), we chose the midpoint of the deuterium measurement period (day 7 of 14).

All mean milk intake volumes of cohorts were individually plotted in a graph and further pooled at every month during the first year of life, together with the available SDs [21,22]. Additional intermediate time points were calculated depending on the relative sample sizes of the cohorts. A non-linear regression was fitted using RStudio (version 2023.12.1+402) to construct a logarithmic function based on the function used in Rios-Leyvraz et al., allowing for the comparison to Yeung et al. or Rios-Leyvraz et al. [7,15].

## 3. Results

A total of 1042 articles from the six databases were derived. After duplicate removal, 395 articles had to be screened (PRISMA flow diagram, Figure 1) [17,18]. After screening of the title and abstract, 96 articles were retained for full-text assessment. Subsequently, 36 articles were excluded, predominantly because the articles did not report human milk intake volume (*n* = 12) or did not report the use of the dose-to-the-mother deuterium oxide dilution method (*n* = 10). This resulted in a total of 60 included articles.

After deciding on the study design and selecting the preferred quality assessment template, the included articles were assessed for quality and risk of bias. Sixteen articles were classified as cohort studies and another thirty-three cross-sectionally studied human milk intake volume (Appendix A). The remaining 11 studies were randomized controlled trials (Appendix A). Related to quality assessment, the mean score on the NOS for cohort studies was 7.6 out of 9 points (good quality), with the lowest score being 6 out of 9 points [19]. Cross-sectional studies obtained a mean score of 5.1 out of 7 points, with the lowest score being 4 out of 7 [19]. All 11 randomized controlled trials were of good quality with an overall low risk of bias [20].

Of all 60 articles included, 10 articles were merged into five datasets. This decision was based on the interpretation that these articles reported on the same cohort and deuterium oxide dilution results. One study cohort was excluded as it only contained infants born as small for gestational age [25]. Two studies reported data as median and interquartile range, disabling further data synthesis [26,27].

The studies were conducted in 34 different countries on six continents (Appendix A): Africa (42.6%), Asia (21.3%), South America (14.8%), North America (13.1%), Europe (4.9%) and Oceania (3.3%). The pooled mean age of mothers (mean ± SD) was 26.8 ± 5.7 years, obtained from studies reporting on this at any point during the study (involving 3842 mothers). Similarly, the pooled mean weight of mothers (mean ± SD) reported at any point during the study was 57.5 ± 10 kg (involving 2595 mothers). Mothers breastfed their infants either exclusively or non-exclusively (predominantly or partially). This vastly differed between cohorts and the infantile postnatal age.

### 3.1. Human Milk Intake Volume—mL/Day

A total of 152 data points were derived, representing the mean human milk intake volume of 5502 exclusively and non-exclusively breastfed infants. For eight data points, we were not able to retrieve the matching sample sizes. The data are visualized in Figure 2, displaying the mean human milk intake volume (mL/day) of cohorts at their respective postnatal age (months). The pooled means and SDs of human milk intake volume were calculated at intermediate time points during the first year of life (Figure 3 and Table 1).

### 3.2. Human Milk Intake Volume—mL/kg/Day

A total of 111 data points were derived, representing the mean human milk intake volume of 3657 exclusively and non-exclusively breastfed infants. For eight data points, we were not able to retrieve the matching sample sizes. The data are visualized in a graph (Figure 4), displaying the mean human milk intake volume (mL/kg/day) of cohorts at their respective postnatal age (months). The pooled means and SDs of human milk intake volume were calculated at intermediate time points of the first year of life (Figure 5 and Table 2).

### 3.3. Function of Human Milk Intake Volume (mL/kg/Day) at Different Postnatal Ages (Days)

The function of pooled mean human milk intake volume (mL/kg/day) at different postnatal ages in days is best described by the following function (Equation (1)), as shown in Figure 6:(1)$$y\;\left(\mathrm{mL}/\mathrm{kg}/\mathrm{day}\right)=149.40024-0.22681\times x-0.31437\times \log\left(x\right)$$

This logarithmic regression has an R-squared value of 0.68, depicted in Figure 6. Here, the current function can be compared to the non-linear regression derived by Yeung et al. and Rios-Leyvraz et al. [7,15].

### 3.4. Human Milk Intake Volume Function (mL/kg/Day) During First 6 Months

In this review, we primarily focused on human milk intake volume during the first 6 months of age. Most of the collected data in this review were from infants up to 6 months of age (predominantly at 6 weeks, 3 and 6 months) and thus matched our primary interest period. The logarithmic regression performed during the primary interest period up to 6 months of age yielded the following function (Equation (2)), as depicted in Figure 7:(2)$$y\;\left(\mathrm{mL}/\mathrm{kg}/\mathrm{day}\right)=127.4485-0.3853\times x+19.4677\times \log\left(x\right)$$
where x = postnatal age in days; this logarithmic regression reproduces an R-squared of 0.54.

### 3.5. Maternal Condition and Intervention Effects on Human Milk Intake Volume

Two African studies reporting on maternal HIV status in relation to human milk intake volume did not show significant differences at both the 1.5- and 6-month measurement periods (Table 3) [28,29]. Furthermore, one of these studies confirmed this non-significance at the 3-, 9- and 12-month measurement periods [28].

A study on the presence of perinatal depression among Pakistani mothers did not show any significant difference in human milk intake volume at 4 months of age either [30]. In contrast, Chilean mothers who smoked (≥4 cigarettes per day) during pregnancy and breastfeeding did show a (*p* < 0.0001) lower human milk intake volume [31].

An intervention of postpartum deworming with Albendazole in Peru showed no significant lower human milk intake volume compared to mothers in the control group [32].

**Table 3 nutrients-16-04205-t003:** Maternal condition and intervention effects on human milk intake volume.

Author	Year	Postnatal Age	Sample Size	Maternal Conditions	Human Milk Intake Volume	*p*-Value
Mulol et al. [28]	2016	6 weeks	21	HIV positive	831 ± 185 g/day	0.06
			24	HIV negative	948 ± 223 g/day	
		3 months	28	HIV positive	899 ± 188 g/day	0.61
			45	HIV negative	925 ± 227 g/day	
		6 months	27	HIV positive	871 ± 293 g/day	0.66
			45	HIV negative	902 ± 286 g/day	
		9 months	24	HIV positive	679 ± 281 g/day	0.33
			43	HIV negative	746 ± 263 g/day	
		12 months	13	HIV positive	755 ± 287 g/day	0.64
			33	HIV negative	713 ± 264 g/day	
Oiye et al. [29]	2023	6 weeks	68	HIV positive	721 ± 111 g/day	0.88
			65	HIV negative	719 ± 121 g/day	
		6 months	60	HIV positive	960 ± 121 g/day	0.91
			62	HIV negative	963 ± 107 g/day	
Rahman et al. [30]	2016	4 months	24	Depressed	89.3 ± 38.1 mL/kg/day	0.57
			31	Not depressed	83.9 ± 29.0 mL/kg/day	
Vio et al. [31]	1991	41 ± 6.7 days	10	Smoking	693 ± 110 g/day	<0.0001
		52 ± 14 days	10	Not smoking	961 ± 120 g/day	
Mofid et al. [32]	2021	1 months	109	Albendazole	756 ± 167 mL/day *	0.471
			90	Placebo	774 ± 170.8 mL/day *	
		6 months	107	Albendazole	903 ± 165.5 mL/day *	0.849
			93	Placebo	908 ± 173.6 mL/day *	

Sample size: number of mother–infant pairs. Human milk intake volume: mean ± standard deviation (standard deviation calculated from provided standard error (*)).

## 4. Discussion

This systematic review reported on the human milk intake volume of approximately 5502 infants being either exclusively or non-exclusively breastfed during the first year of life. Of the 60 articles included, most cohort studies were of good quality (≥7 points on the NOS), and all included randomized controlled trials had a low risk of bias [19,20].

The pooled means of human milk intake volume (mL/day) at different postnatal ages seemed to reach a convex plateau phase, with the highest pooled mean observed at 5 months old, measuring 882.4 ± 181 mL/day. When the available data were converted to mL/kg/day, the trendline followed a downward slope. 

While the current regression model performed well overall, our function deviated noticeably from that of Yeung et al. during the first few weeks [7]. The current function described the maximal value in the first week of life, while the human milk intake volumes reported by Yeung et al. were considerably lower and reached their maximal value at about 2 weeks of age [7]. These lower intakes are expected in real-world practice, as colostrum intake volume in the very first few days after birth is low and rises rapidly over the subsequent days [33,34]. This deviation can be attributed to a limitation of the deuterium oxide dilution method, namely its 14-day runtime during which multiple saliva (or urine) samples have to be taken [8]. In contrast, the test-weighing method has to be performed within a minimum of 24 h and can be applied from early neonatal life onwards [35]. Consequently, the test-weighing method is preferred during the first few postnatal week(s), which explains the relative under-representation of data using the deuterium oxide dilution method in the first few weeks of life, thus resulting in a suboptimal regression analysis [11]. Despite the 14-day length of the deuterium oxide dilution method, it was proven to be a usable method in early infancy in one study (starting within the first 48 h) [36]. Therefore, the current function is not suitable for the first few weeks of life; however, our function during the first 6 months of life presents this aspect of early infancy more accurately.

Overall, our systematic review confirms the presence of a high-risk period for medication toxicity due to high human milk intake volume in early infancy [7]. As can be observed in Figure 6, the maximal intake expressed in mL/kg/day occurs in the age window up to 2.5 months. This window has a specific risk as this higher intake holds a risk for medicine accumulation which co-occurs with the still low drug clearance capacity of the infant [7,37].

In essence, our model converged and surpassed the function described by Yeung et al. after approximately 2 to 3 months [7]. Despite including both exclusively and non-exclusively breastfed infants in our model, equal or higher intakes were observed compared to the exclusively breastfed infants included in the study of Yeung et al. [7]. However, lower intakes were expected because the included infants were not exclusively breastfed. As exemplified by the few studies (Appendix A) that did report separate intakes (per day) for infants with comparable weights, the exclusively breastfed infants showed significantly higher intakes than non-exclusively breastfed infants [38,39,40,41].

Our model was not able to account for the breastfeeding exclusivity rate since most studies included both exclusively and non-exclusively breastfed infants, often without separate reporting of their human milk intake volume data. Yet, a convergence was observed between our function and that of Yeung et al. which could be explained by the findings of the recent systematic review conducted by Rios-Leyvraz et al. [7,15]. They reported no significantly different intakes between the dose-to-the-mother deuterium oxide dilution method and the test-weighing method corrected for insensible water loss [15]. However, it could be assumed that studies using the infantile test-weighing method seldomly correct for insensible water loss, explaining the potential underestimation of human milk intake volume in the study of Yeung et al. [7,15,42].

Compared to the overall function on all data of Rios-Leyvraz et al., our current function systematically reported higher intakes [15]. The lower intake of their review might be explained by an over-representation of test-weighing data, as lower test-weighing results are described compared to deuterium dilution results [34]. Furthermore, test-weighing was primarily self-managed by mothers (52%) or the executor was unclear (20%), possibly contributing to more variation in practice and methods of weighing [15]. Lastly, differences in ethnicities and feeding habits might help explain the discrepancy between different reviews [43].

This review expresses the presence of a high-risk period in early infancy concerning the transfer of maternal pharmacotherapy though human milk to the infant. Moreover, it questions the function described by Yeung et al. after 2 months of age because their model may underestimate the actual human milk intake volume due to limited correction of insensible water loss [7]. This underestimation may be even more present in the model of Rios-Leyvraz et al., when compared to our model [15]. This has important implications for PBPK simulations as the risk estimation can be higher during later infancy. After 6 months of age, some authors stated that the deuterium oxide dilution method is not applicable because of varying breastfeeding practices [9]. However, this was less of a concern as our study primarily focused on human milk intake volume during the first 6 months of age, which was more accurately presented by Equation (2).

Among the maternal conditions and interventions relevant as potential PBPK study topics, only one factor showed a significant impact on infantile human milk intake volume. Specifically, maternal smoking (≥4 cigarettes per day) during pregnancy and breastfeeding was found to negatively influence human milk intake volume in infants [31]. Consequently, when investigating the risk of maternal smoking on infants using PBPK modeling, it may be appropriate to apply a correction for this lower milk intake volume. As the other maternal conditions and interventions showed no relevant differences, the need for correction is not required, thereby further validating the applicability of our model for general PBPK modeling purposes. Importantly, this includes maternal HIV infection (Table 3), as this suggests that the reference values on human milk intake volume can also be used in this specific scenario.

To our knowledge, this systematic review represents the largest study to date that pools human milk intake volume data obtained solely using the (dose-to-the mother) deuterium oxide dilution method, considerably larger than a previous review by da Costa et al. from 2010 [14]. Unlike da Costa et al., this current model reported human milk intake volumes relative to changing infant weight, providing more informative data for PBPK modeling [14]. Furthermore, our review illustrates the increased use of the deuterium oxide dilution method for estimating human milk intake volume, with a gradual increase in its use in the industrialized world in recent years [35].

One limitation of both the deuterium oxide dilution method and our systematic review is the relatively limited availability of data from industrialized or European countries, which could impact the input for PBPK models. To illustrate this, only 4.9% of the studies included in our review reported data from Europe. Notably, the study by Rios-Leyvraz et al. using both test-weighing and deuterium oxide dilution data reported significantly different intake volumes across continents [15]. Since our model did not require exclusive breastfeeding of infants, it can provide a better representation of actual intakes in a population. However, it is important to recognize that breastfeeding practices also vary depending on the country of study [44]. Moreover, the maternal characteristics differed across populations in the studies, with an overall pooled maternal age of 26.8 ± 5.7 years and a pooled weight of 57.5 ± 10 kg.

Another limitation of our study is the sparse availability of data regarding human milk intake volume expressed in mL/kg/day. To address this, the mean human milk intake volume was converted from mL/day to mL/kg/day using the mean infant weights. This conversion provided highly informative data and allowed for comparison with the function of Yeung et al. or Rios-Leyvraz et al. [7,15]. As articles seldomly reported intakes directly in mL/kg/day, it can be concluded that these converted data only represent a minor limitation associated with the use of the deuterium oxide dilution method.

## 5. Conclusions

This systematic review highlights the value of using the (dose-to-the-mother) deuterium oxide dilution method for the estimation of human milk intake volume to generate a function over postnatal age. These data are time (postnatal age)-sensitive, as breastfeeding volumes may negatively affect insensible water loss compared to test-weighing. Our findings have important implications for PBPK simulations, as risk estimation can be higher during later infancy. However, our model solely using data from the deuterium oxide dilution method may also underestimate the risk for exclusively breastfed infants due to the inclusion of non-exclusively breastfed infants. Furthermore, this review highlights the lack of data using the deuterium oxide dilution method, especially in the early postnatal period before 1 month of age and in the industrialized world or Europe. This review also urges for more data to be presented relative to infantile weight (mL/kg/day) and separate reporting of breastfeeding exclusivity rates, as this is most informative for PBPK predictions. Based on the findings of this review, future mother–infant PBPK models can be enhanced to further inform healthcare decisions and improve outcomes for both mother and infant.

## Figures and Tables

**Figure 1 nutrients-16-04205-f001:**
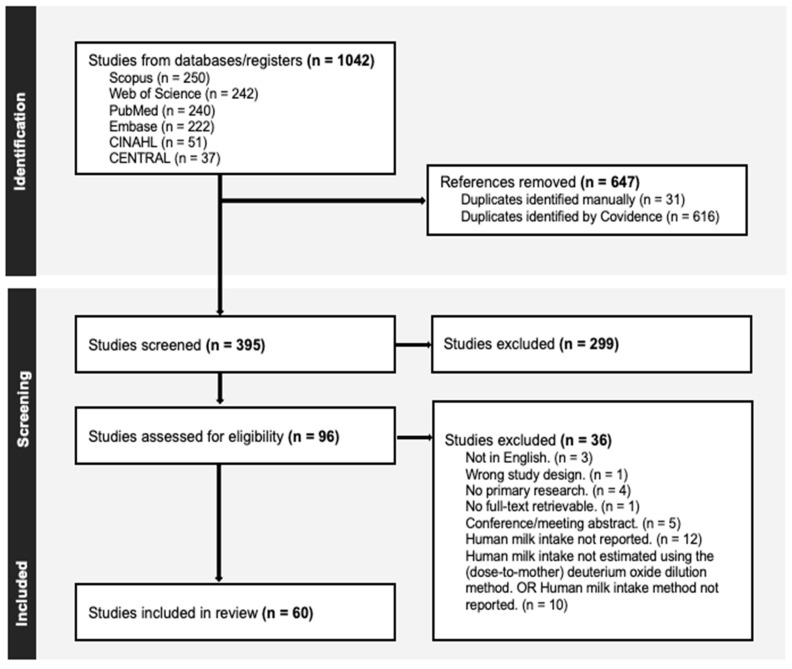
A visual representation of the selection process of articles in the form of a PRISMA flow diagram [17,18]. (*n*) represents the number of studies in the different stages of selection.

**Figure 2 nutrients-16-04205-f002:**
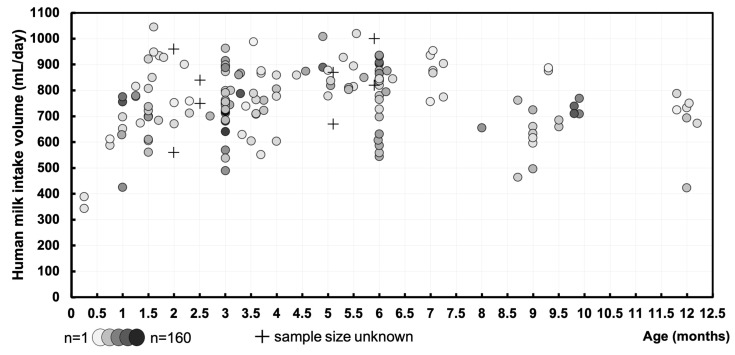
Human milk intake volume (mL/day) at different postnatal ages (months) during the first year of life of exclusively and non-exclusively breastfed infants. Markers represent individual cohorts at the measurement periods, where darker circles correspond to larger sample sizes (minimum *n* = 1/maximum *n* = 160). Where no information on the sample sizes is known, these are depicted as crosses.

**Figure 3 nutrients-16-04205-f003:**
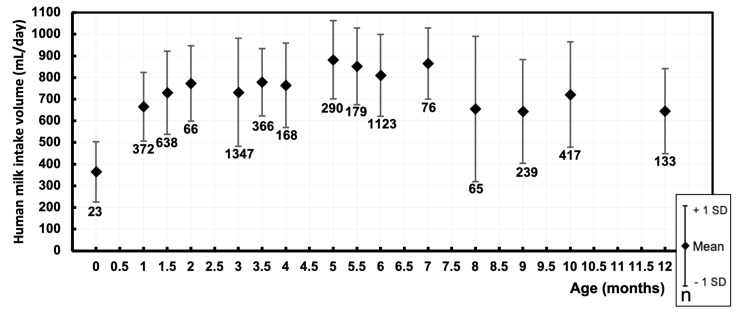
Pooled mean human milk intake volume (mL/day) at different postnatal ages (months) during the first year of life of exclusively and non-exclusively breastfed infants. Markers represent the pooled mean human milk intake volume and the range of estimated standard deviations at the respective postnatal age. Sample sizes are depicted under the markers.

**Figure 4 nutrients-16-04205-f004:**
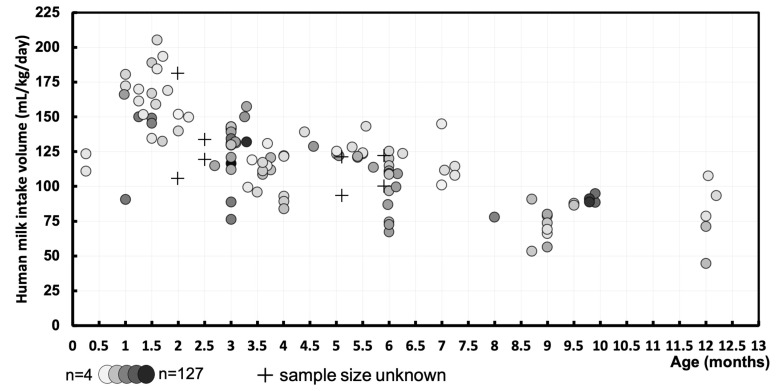
Human milk intake volume (mL/kg/day) at different postnatal ages (months) during the first year of life of exclusively and non-exclusively breastfed infants. Markers represent individual cohorts at the measurement periods, where darker circles correspond to larger sample sizes (minimum *n* = 4/maximum *n* = 127). Where no information on the sample sizes is known, these are depicted as crosses.

**Figure 5 nutrients-16-04205-f005:**
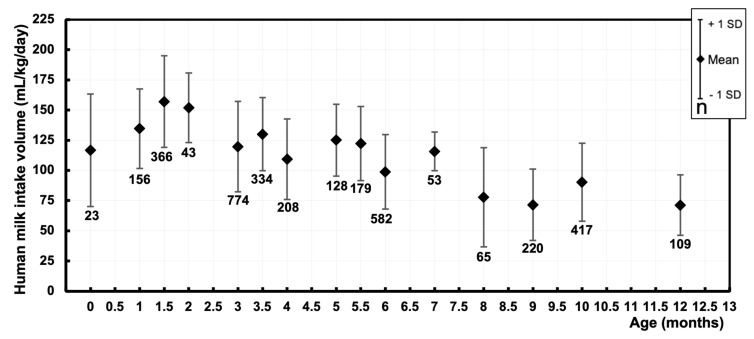
Pooled mean human milk intake volume (mL/kg/day) at different postnatal ages (months) during the first year of life of exclusively and non-exclusively breastfed infants. Markers represent the pooled mean human milk intake volume and the range of estimated standard deviations at the respective postnatal age. Sample sizes are depicted under the markers.

**Figure 6 nutrients-16-04205-f006:**
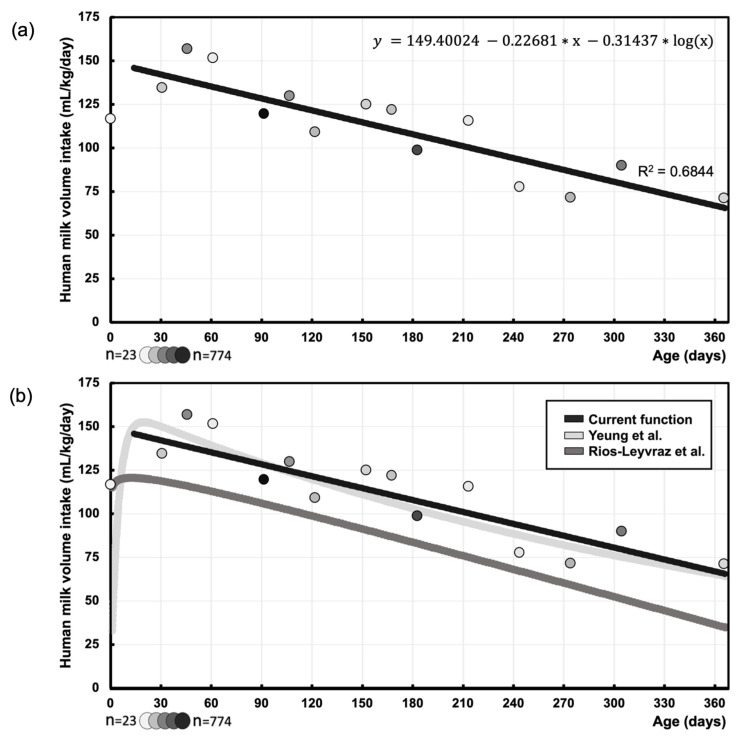
(**a**) The current function of pooled mean human milk intake volume (mL/kg/day) at different postnatal ages in days, with the corresponding formula and R-squared value. Circle markers represent the pooled means, with darker circles corresponding to larger sample sizes (minimum *n* = 23, maximum *n* = 774). (**b**) The current function, next to the functions by Yeung et al. and Rios-Leyvraz et al. [7,15].

**Figure 7 nutrients-16-04205-f007:**
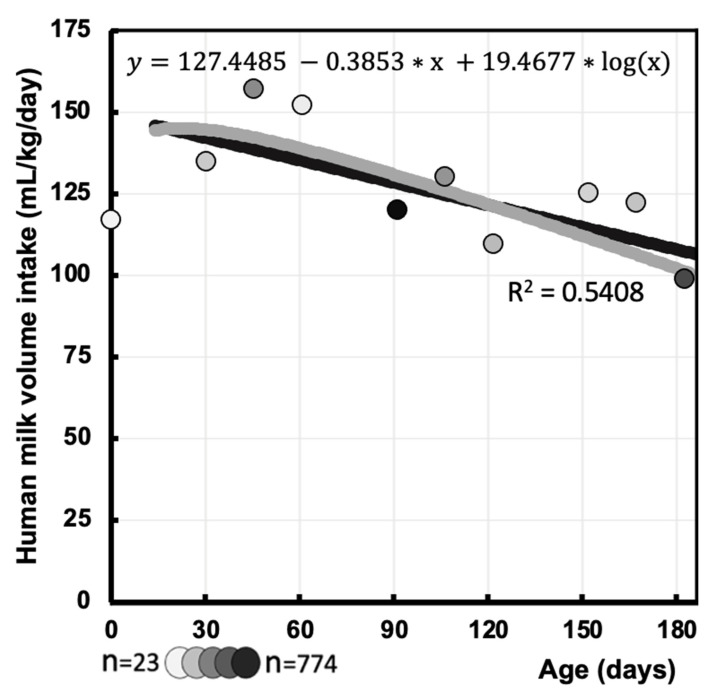
The gray function describing human milk intake volume (mL/kg/day) at different postnatal ages in days during the first 6 months of age. The black line represents our function over the first year of life, whereas circle markers represent the pooled means, with darker circles corresponding to larger sample sizes (minimum *n* = 23, maximum *n* = 774).

**Table 1 nutrients-16-04205-t001:** Pooled mean human milk intake volume and SD (mL/day) during first year of life of exclusively and non-exclusively breastfed infants.

Postnatal Age (Months)	Pooled Sample Size (*n*)	Pooled Mean (mL/Day)	Pooled SD (mL/Day)
0 [0.01–0.49]	23	364.9	139
1 [0.50–1.24]	372	665.6	158.3
1.5 [1.25–1.74]	638	729.9	191.9
2 [1.75–2.49]	66	773.4	174.4
3 [2.50–3.24]	1347	732	249.4
3.5 [3.25–3.74]	366	778.6	155.2
4 [3.75–4.49]	168	764.7	194.8
5 [4.50–5.24]	290	882.4	181
5.5 [5.25–5.74]	179	852.5	177
6 [5.75–6.49]	1123	809.9	189
7 [6.50–7.49]	76	864.8	164.9
8 [7.50–8.49]	65	655	336
9 [8.50–9.49]	239	643.5	239.2
10 [9.50–10.49]	417	721.7	243
12 [11.50–12.49]	133	645	197

**Table 2 nutrients-16-04205-t002:** Pooled mean human milk intake volume and SD (mL/kg/day) during first year of life of exclusively and non-exclusively breastfed infants.

Postnatal Age (Months)	Pooled Sample Size	Pooled Mean (mL/kg/Day)	Pooled SD (mL/kg/Day)
0 [0.01–0.49]	23	116.8	46.6
1 [0.50–1.24]	156	134.6	32.9
1.5 [1.25–1.74]	366	157	38
2 [1.75–2.49]	43	151.9	28.9
3 [2.50–3.24]	774	119.7	37.3
3.5 [3.25–3.74]	334	130.1	30.3
4 [3.75–4.49]	208	109.4	33.4
5 [4.50–5.24]	128	122.2	30.7
5.5 [5.25–5.74]	179	122.2	30.7
6 [5.75–6.49]	582	98.8	30.8
7 [6.50–7.49]	53	115.8	16
8 [7.50–8.49]	65	77.8	41
9 [8.50–9.49]	220	71.5	29.5
10 [9.50–10.49]	417	90.2	32.3
12 [11.50–12.49]	109	71.2	25

## Data Availability

The corresponding author can be contacted to share the search results, and data will be shared upon reasonable request.

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
