# Peer review of "The Deuterium Oxide Dilution Method to Quantify Human Milk Intake Volume of Infants: A Systematic Review—A Contribution from the ConcePTION Project"

_nutrients, 2024, doi:10.3390/nu16234205_

Round 1

Reviewer 1 Report

Comments and Suggestions for Authors

Manuscript #3347538 entitled " Deuterium oxide dilution method to quantify human milk intake volume of infants: systematic review – A contribution from the ConcePTION project" 

Thank you for giving an opportunity to review this interesting paper. This systematic review focuses on important subject of measuring and potentially referencing human milk intake with validated method (deuterium oxide dilution) and potential prediction of medication transfer through breast milk. Below are comments and suggestions for changes that may improve the manuscript.

Comments to the Authors

Overall, this is a very significant topic. And paper is very well written and presented. The systematic review was registered with PROSPERO and is reported in accordance with PRISMA recommendations.

1. Introduction

Please mention technical/cost limitations, measurements errors and compliance issues with deuterium oxide dilution in addition to test-weighing method issues. Some of them are described in Discussion. Whilst it sufficed for this particular study, deuterium oxide dilution is not accounting for commercial formula or donor milk consumed by the infant and poses timing challenges with expressed breast milk feeds, thus may not be optimal for some infant nutrition and growth studies.

Scanlon et al., 2002. Assessment of infant feeding: the validity of measuring milk intake.

3. Results

In Figures 2, 4, 6 and 7, I suggest indicating sample size ranges for each circle in the figure label (or at least in the figure legend) to remove the need of calculating/estimating it. There is enough space for that.

Line 294 – no need for word ‘significant’ next to p-value, just ‘lower’ indicates it is significant.

Line 296 – ‘compared to mothers from control’ group instead of ‘control mothers’ – people first approach.

It seems the bias assessment results are not reported, with the first mentioning being in Discussion (Line 306). I suggest having a Results section with a short bias assessment summary, even keeping the tables as Supplementary.

4. Discussion and 5. Conclusions

Lines 365-367 – “Thus, the lower intakes from non-exclusively breastfed infants in our model were balanced by the possible underestimation of human milk intake volume using the test-weighing method.” – I cannot quite agree with this statement, these data cannot be compared and/or matched based on methods’ imperfections, please rephrase/remove. Further, data are time (postpartum/age) sensitive, as breastfeed volumes and potentially feeding frequency may negatively affect insensible water loss with test-weighing, this is mentioned in Conclusions though. I suggest discussing it more clear before the Conclusions and make a statement in Conclusions on when deuterium oxide dilution method can be used reliably, e.g., between 1 and 6 months of age.

Kind regards,

Author Response

Overall, this is a very significant topic. And paper is very well written and presented. The systematic review was registered with PROSPERO and is reported in accordance with PRISMA recommendations.

We thank the reviewer of the overall very positive and supportive assessment of the paper, as we have indeed tried to respect all relevant guidelines.  

Please mention technical/cost limitations, measurements errors and compliance issues with deuterium oxide dilution in addition to test-weighing method issues. Some of them are described in Discussion. Whilst it sufficed for this particular study, deuterium oxide dilution is not accounting for commercial formula or donor milk consumed by the infant and poses timing challenges with expressed breast milk feeds, thus may not be optimal for some infant nutrition and growth studies. Scanlon et al., 2002. Assessment of infant feeding: the validity of measuring milk intake.

While this was not the 'main' topic of the current systematic review, we agree that - as with any technique, there are limitations. We have therefore added some sentences and the reference mentioned. 

In Figures 2, 4, 6 and 7, I suggest indicating sample size ranges for each circle in the figure label (or at least in the figure legend) to remove the need of calculating/estimating it. There is enough space for that.

We felt that this would overload the figures, while the information on the pooled sample size is provided in the table 1. We therefore hope that the reviewer agrees on our approach taken.

Line 294 – no need for word ‘significant’ next to p-value, just ‘lower’ indicates it is significant, and Line 296 – ‘compared to mothers from control’ group instead of ‘control mothers’ – people first approach.

We have implemented these suggestions. 

It seems the bias assessment results are not reported, with the first mentioning being in Discussion (Line 306). I suggest having a Results section with a short bias assessment summary, even keeping the tables as Supplementary.

We felt tthat this information was already provided, but perhaps this was not yet sufficiently well stressed in the results section. We have added some info on this in the results section, and rephrased this (bold, italics).

Related to quality assessment,the mean score on the NOS-scale for cohort studies was 7.6 out of 9 points (good quality), with the lowest score being 6 out of 9 points [18]. Cross-sectional studies obtained a mean score of 5.1 out of 7 points, with the lowest score being 4 out of 7 [18]. All 11 randomized controlled trails were of good quality with an overall low risk of bias [19].

. Discussion and 5. Conclusions

Lines 365-367 – “Thus, the lower intakes from non-exclusively breastfed infants in our model were balanced by the possible underestimation of human milk intake volume using the test-weighing method.” – I cannot quite agree with this statement, these data cannot be compared and/or matched based on methods’ imperfections, please rephrase/remove.

the statement has been removed. 

Further, data are time (postpartum/age) sensitive, as breastfeed volumes and potentially feeding frequency may negatively affect insensible water loss with test-weighing, this is mentioned in Conclusions though. I suggest discussing it more clear before the Conclusions and make a statement in Conclusions on when deuterium oxide dilution method can be used reliably, e.g., between 1 and 6 months of age.

We have moved the commented text section to the discussion section, and remove the text in the conclusions section.

While we understand the comment of the reviewer, our initial intention was not to assess the reliability of the deuterium technique, as others have done this previously. 

Reviewer 2 Report

Comments and Suggestions for Authors

The authors have undertaken an interesting topic and made a commendable effort to systematically collect available data from clinical studies that report human milk volumes solely using the deuterium oxide dilution method at different postnatal ages. Overall, the manuscript is well-written and I recommend it for publication in its current form.

Author Response

Comments

The authors have undertaken an interesting topic and made a commendable effort to systematically collect available data from clinical studies that report human milk volumes solely using the deuterium oxide dilution method at different postnatal ages. Overall, the manuscript is well-written and I recommend it for publication in its current form.

We thank the reviewer for the overall very supportive assessment. We have also considered the annotated pdf as provided. We do prefer to keep both the figures and the tables in the paper, so that the data are available to either assess visually, or to extract (table) for other readers. We have adapted the text on 'young infants' in line 75, as requested.